# Transcriptional Networks of Microglia in Alzheimer’s Disease and Insights into Pathogenesis

**DOI:** 10.3390/genes10100798

**Published:** 2019-10-12

**Authors:** Gabriel Chew, Enrico Petretto

**Affiliations:** Programme in Cardiovascular and Metabolic Disorders, Duke-NUS Medical School, 8 College Road, 69857 Singapore, Singapore; e0008504@u.nus.edu

**Keywords:** Alzheimer’s disease, microglia, gene networks, transcriptomics, microarray, RNA-seq, single-cell RNA sequencing

## Abstract

Microglia, the main immune cells of the central nervous system, are increasingly implicated in Alzheimer’s disease (AD). Manifold transcriptomic studies in the brain have not only highlighted microglia’s role in AD pathogenesis, but also mapped crucial pathological processes and identified new therapeutic targets. An important component of many of these transcriptomic studies is the investigation of gene expression networks in AD brain, which has provided important new insights into how coordinated gene regulatory programs in microglia (and other cell types) underlie AD pathogenesis. Given the rapid technological advancements in transcriptional profiling, spanning from microarrays to single-cell RNA sequencing (scRNA-seq), tools used for mapping gene expression networks have evolved to keep pace with the unique features of each transcriptomic platform. In this article, we review the trajectory of transcriptomic network analyses in AD from brain to microglia, highlighting the corresponding methodological developments. Lastly, we discuss examples of how transcriptional network analysis provides new insights into AD mechanisms and pathogenesis.

## 1. Introduction: The Role of Microglia in Alzheimer’s Disease

Alzheimer’s disease (AD) is a devastating neurodegenerative disease associated with cognitive decline, memory deficits, and loss of executive function while accounting for 50% to 75% of dementia cases [1]. It is increasingly prevalent and typified by the accumulation of extracellular amyloid structures and intracellular tau fibrils [2,3]. Neuropathologically, AD is associated with accumulation of amyloid structures starting in the neocortex before spreading to the allocortex and eventually the cerebellum [4]. Despite having known pathological correlates, AD is a complex multifactorial disease influenced by cardiovascular risk factors [5] and biopsychosocial variables such as education attainment [6], socio-economic status [7], and level of physical activity [8,9]. Nevertheless, a majority of basic research, translational, and clinical efforts hinge on the amyloid cascade hypothesis [10,11] which posits that amyloid build-up in vulnerable brain regions triggers a wave of downstream molecular events, ultimately leading to the manifestation of AD symptoms. Such events include the recruitment of microglia, further seeding of amyloid, neurofibrillary accumulation, and neuronal loss [12]. Therefore, withholding the amyloid cascade is widely adopted as a pharmaceutical strategy [3]. Unsurprisingly, microglia, the main phagocytes of the central nervous system (CNS) [13], have been intensively studied in the context of AD pathogenesis [14,15,16,17].

Microglia are the immune sentinels of the CNS, and they maintain homeostasis by clearing debris, pruning synapses, and interacting with other brain cell types [13]. In addition to AD, microglia’s neuroinflammatory role has been implicated in Parkinson’s disease [18], multiple sclerosis [19], and stroke [20]. However, the specific role of microglia in AD pathogenesis remains to be mechanistically resolved [21]. On one hand, microglia phagocytose amyloid and protect neurons from amyloid toxicity [18,22]. On the other, they release pro-inflammatory mediators which cause neuronal damage [23] and can potentially exacerbate inflammation by interacting with other cell types [24]. The latter could depend on the chronicity of inflammation [25]. During acute inflammation instigated by amyloid build-up, microglia serve a beneficial role by surrounding neurons and limiting neuronal exposure to amyloid [18,22]. Normally, acute inflammation is followed by a resolution phase through microglia’s anti-inflammatory action [26]. In AD, amyloid build-up remains, resulting in continuous insult to the brain parenchyma [26]. This chronic inflammation driven by microglia’s release of cytokines and reactive oxygen species (ROS) causes loss of synapses through dysfunctional microglia pruning [27]. Inflammation can also be modulated by age and, compared to young brains, old brains have leaky blood–brain barrier (BBB) which allow infiltration of pro-inflammatory monocytes [26]. In old brains, there is also a higher number of activated microglia due to chronic inflammation, further supporting that age is a major risk factor for AD [26]. In addition, Cribbs et al. found that immune-related genes are dysregulated with aging and AD [28], while co-culturing old microglia with media from young microglia has been found to be potentially sufficient for re-establishing normal microglia proliferation and function [29]. 

Microglia’s role in AD pathogenesis is highly complex and most likely depends on contextual factors such as disease chronicity of inflammation [25], cognitive reserve [30,31], and presence of comorbidities [26,27]. Despite this complexity, there are clear genetic and molecular underpinnings for microglia’s key role in AD. For example, the Triggering Receptor Expressed on Myeloid Cells 2 (*TREM2*) gene has been identified as a key player for microglia function in AD pathology [32,33,34], where *TREM2* loss of function is associated with increased amyloid burden [35] and exacerbated neuronal loss [36]. Genome-Wide Association Studies (GWAS) for late-onset Alzheimer’s disease (LOAD) identified the *TREM2* missense variant R47H [37,38,39], which was also associated with impaired microglia function [40,41]. Apolipoprotein E (APOE), which is the most common AD risk gene, is also related to increased immune cell activation [26]. In particular, the gene isoform associated with the greatest AD risk, *APOE4*, is associated with greater inflammation in response to a pro-inflammatory stimulus [42]. Significantly, the genetic/functional interaction between *APOE* and *TREM2* genes is needed for microglia to switch from a “homeostatic” state to a “neurodegenerative” state [26]. A role for gene–gene interactions in AD is further supported by Tansey et al. who found that genetic risk for AD is enriched in microglia gene networks, i.e., multiple interacting genes in microglia [43]. In particular, DNA binding motifs for *SPI1* and *MEF2*, which are both important regulators of microglia function, are associated with several AD risk genes that interact within a complex gene network [43]. Therefore, gene network analysis of microglia in homeostasis and disease has been proposed to clarify the functional role and specific contribution of multiple gene targets to AD pathogenesis.

## 2. Transcriptomic Networks to Understand Functional Gene-Gene Interactions in AD

### 2.1. Primers of Transcriptional Gene Networks

In its essence, network biology provides a means to describe a biological process as the complex interplay between underlying cellular events at different levels, including (but not limited to) genomics, transcriptomic, and proteomic. This kind of description offers several advantages over reductionist approaches. First, it allows us to integrate at the systems-level the high-content “*-omics*” data, which are now increasingly accessible to the scientific community, to provide high-level mechanistic understanding of genotype-phenotype relationships in disease [44]. Second, network biology provides a platform for not only “describing” a phenomenon, which traditional molecular biology does well, but also “predicting” (and sometimes quantifying) systems-wise outcomes resulting from different perturbations, e.g., genetic mutations and inflammatory insult [44]. In the context of transcriptional data, analysis of gene networks not only identifies concurrently expressed (or co-expressed) genes (i.e., a gene co-expression network), but can also aid the biological interpretation of apparently disconnected genes dysregulated in disease, which are typically detected as differentially expressed (DE) between disease and control samples. Individual DE genes can form a coherent gene network, which can be functionally annotated (e.g., using Gene Ontology (GO), http://geneontology.org/) to inform specific biological processes and molecular pathways dysregulated in disease.

Generally speaking, a gene network consists of a graph representation of genes (represented as nodes) and their functional interactions (represented as edges). However, it is important to distinguish a gene network from a gene “signature” (also referred to as transcriptional signature). A gene signature refers to a combined group of genes with a characteristic pattern of gene expression in a cell/tissue—for example a set of dysregulated genes (e.g., identified by differential expression analysis between disease and healthy states). In contrast, a network describes a set of interconnected genes (which may or may not be DE) alongside their functional interaction by means of some measure of gene–gene connectivity. This gene–gene connectivity can be quantified in a population sample based on different approaches (Figure 1), for example these include correlation analysis, mutual information (MI) [45,46,47], Bayesian methods [48], random forest [49], and regression-based methods [50]. The relative advantages and disadvantages of each approach are reviewed elsewhere [51,52].

In transcriptomic network analysis, the gene–gene connectivity is typically quantified as gene co-expression. Specifically, co-expression quantifies the extent to which two genes are expressed (or silenced) together in a given context (e.g., tissue, disease, during development or in response to molecular/environmental perturbation). In its more simple assessment, gene co-expression can be calculated using either Pearson or Spearman correlation of the genes’ expression profile in the population sample (e.g., in a cohort of AD patient brains). Therefore, correlation-based co-expression networks (also known as “relevance networks”) result in undirected network graphs, which simply represent gene–gene connections and where the specific action (activation or repression) of a gene on another is not assessed and represented in the network [53]. A related but distinct approach used to assess gene co-expression is Graphical Gaussian Models (GGMs). GGMs employ partial correlations, which measure the extent of linearity between two genes’ expression profiles in the context when the variability attributed to a third gene(s) expression profile is removed [54]. Briefly, GGMs are constructed by estimation of the inverse covariance matrix [55], which describes the “conditional dependence” between every two gene expression profiles. This enables the identification of direct gene–gene relationships or “conditional independence networks”, which have been found to reflect better biological network structures compared to “relevance networks” [56]. In the next section, we will elaborate on co-expression network analysis in disease and review gene networks studies in AD, mentioning a few seminal studies based on gene signature analysis.

### 2.2. From Gene Co-Expression to Regulatory Networks in Disease

In the context of transcriptional gene network analysis, it is also important to distinguish “co-expression” from “co-regulation” because co-expressed genes are neither co-regulated nor necessarily share the same regulatory mechanism. For example, using over 600 microarray experiments on *S. cerevisiae*, Allocco et al. showed that the correlation (*r*) between any two genes’ expression profile must be over *r* = 0.84 in order to have a 50% chance of sharing the same regulation by a given transcription factor [57]. Thus, beyond identifying strong correlation between genes’ expression profiles, construction of co-regulation networks requires incorporation of additional regulatory information, which can be provided, for instance, by transcription factor (TF) binding analysis either using in silico databases [58], or including splicing information [59] or chromatin immunoprecipitation (ChIP) data [60]. For example, Ding et al. incorporated protein interactions and TF binding motif data to construct gene regulatory networks, identifying ATF1 (Activating Transcription Factor 1), which regulates microglia anti-inflammatory action in AD through the cell surface marker CXCR4 [61,62]. To infer gene co-regulation networks, Vargas et al. employed Algorithm for the Reconstruction of Accurate Cellular Networks (ARACNE), which not only infers gene networks but also identifies targets of transcription factors [63,64]. Rather than using correlation-based metrics, ARACNE utilizes MI which informs on how much information one variable (e.g., gene expression profile) carries with regards to another variable [63]. Specifically, MI might be advantageous in network inference over correlation-based metrics because it does not assume linear interactions and indeed can capture nonlinear relationships between genes [45], and it is less influenced by experimental errors [65].

Another important nuance is that gene co-expression alone does not necessarily capture changes in the pathological process, highlighting the utility of differential co-expression analysis in disease. Generally speaking, differential co-expression identifies discrete gene sets whose co-expression patterns vary (or are altered) across conditions, e.g., between disease and healthy states. There are three levels of differential co-expression analysis, which relate to changes to the network topology, gene module, and specific gene-pairs [66]. Network topology embraces key global measures of the network structure, such as the node degree distribution, network clustering coefficient, and average path length [66]. For example, the node degree distribution is the number of edges attached to a node, reflecting a node’s degree of connectivity [66]. Gene modules refers to coherent sets of genes, typically detected to be co-expressed through clustering the genes’ expression profile and then applying some thresholding procedure (for instance, thresholding the whole gene expression correlation matrix to extract smaller gene modules therein). Lastly, gene-pair level analysis zoom into specific gene–gene relationship and corresponding molecular mechanism, e.g., TF and its target genes. Practically, for each level previously described, differential co-expression accounts for (and test) two possible scenarios: (i) Genes are similarly co-expressed but DE between disease and healthy states, (ii) genes are differentially co-expressed but not DE between disease and healthy states. One commonly used approach for the module level analysis, DiffCoEx, for example, measures both intra-module and inter-module differential co-expression [67], while DICER incorporates a probabilistic score for identification of differentially co-expressed gene modules in case-control gene expression data [68].

These approaches for differential co-expression analysis can allow the identification of gene networks or gene modules that are perturbed in disease (Figure 1). For example, Wang and Liu [69] performed differential co-expression analysis in 6 brain regions—entorhinal cortex, hippocampus, medial temporal gyrus, posterior cingulate cortex, superior frontal gyrus, and primary visual cortex, identifying a gene cluster with *NPIPA1* (Nuclear pore complex-interacting protein family member A1) as a “hub gene”. A “hub gene” is a central node in the network with a significantly higher number of connected edges compared to other nodes [70]. Biologically, a “hub gene” could represent a TF or a key signaling molecule with many downstream regulated genes. Similarly, Meng and Mei performed differential co-expression analysis on 1,667 human brain samples to uncover a core network of 97 dysregulated genes in AD, including transcription factors FLI1, NOTCH2, and STAT3 [71]. Notably, FLI1 is known to interact with RUNX1, which is a master regulator for an age-dependent microglia gene co-expression module [72], while NOTCH2 and STAT3 are both involved in microglia activation [73,74]. Therefore, differential co-expression in the brain can unmask gene networks and key “hub genes” therein, such as TFs, which are not only associated with the disease state but also perturbed along the AD trajectory.

## 3. Advancements in Transcriptomics and Gene Network Analysis

### 3.1. Assessing the Transcriptome

The central dogma of molecular genetics posits that RNA serves a messenger role between DNA sequence and protein function/activity, describing the flow of biological information encoded in the DNA from the nucleus to the cytoplasm and/or extracellular space through transcription. Therefore, transcript information (i.e., transcript relative abundance, dysregulation, and localization) can be used to inform cellular processes, feedback mechanisms to external stimuli, and pathological changes, providing insights into the biological state of the cell or tissue sample. Transcriptomic studies of multiple transcripts started with Sanger sequencing techniques such as expressed sequence tags (ESTs) and serial and cap analysis of gene expression (SAGE/CAGE) [75]. In 1995, the novel combination of sequencing, nanofabrication, and fluorescent tagging, gave way to develop the first microarray chip which utilize probes for the corresponding nucleotide sequences, subsequent hybridization, and measurement of fluorescence, which enabled the quantification of gene expression for thousands of genes simultaneously [75,76]. Subsequently, development of higher throughput sequencing methods, namely massive parallel sequencing, also known as deep sequencing or Next Generation Sequencing (NGS), set the stage for RNA-sequencing (RNA-seq), which are able to capture multiple transcripts concurrently without the use of pre-defined reference probes [75].

### 3.2. Microarray-Based Network Analyses in AD

Despite important limitations, such as low dynamic range, and need for a priori knowledge of reference transcript sequences [75], microarrays can provide useful transcriptomic information in AD. In 2004, Blalock et al. studied hippocampal gene expression using 31 separate microarrays for each patient’s tissue sample [77]. This adequately powered study leveraged correlation-based gene network analysis to identify cell growth, differentiation, and tumor suppressors processes to be dysregulated in early AD pathogenesis. Several microarray-based studies have been also utilized to disentangle gene networks in various brain regions, such as the hippocampus [77,78,79,80], superior frontal gyrus [80,81], and the entorhinal cortex [80], highlighting additional biological processes, e.g., inflammation, calcium signaling, and cell proliferation [77]. Most of these pioneer studies used traditional methods for inferring gene–gene interactions, ranging from cluster analysis [78] to gene network topology analysis [82].

In rodents, Uddin and Singh conducted a meta-analysis in the context of age-associated spatial learning impairment (ASLI) [83], a process which is associated with early AD [84] and hippocampal volume loss in mice, and with the AD risk gene isoform *Apoe4* [85]. ASLI-associated gene networks were derived using Ingenuity Pathway Analysis (IPA) software (http://www.ingenuity.com), which allowed to uncover biological processes such as cell signaling, neuronal growth, and synaptic processes in rat learning impairment [83]. In a subsequent study, Uddin and Singh compared gene networks between young (learning unimpaired) and aged (predominantly learning impaired) brains [86], using Weighted Gene Co-expression Network Analysis (WGCNA) [87]—a widely used method to identify gene co-expression patterns, which are summarized as “modules” of highly correlated genes. Within the inferred modules, “hub genes” such as *Camk1g*, *Cdk5r1*, *Dlg3*, *Kcnab2*, and *Mapk1* were identified, and these have been found to be associated with AD, ion signaling, learning, and memory [86]. Thus, multiple human and animal microarray studies consistently highlighted relevant biological processes such as inflammation, neuronal signaling, and developmental processes as important themes associated with AD and comorbidities.

WGCNA was also used in an early seminal microarray co-expression study of the CA1 hippocampal region for 31 individuals ranging from healthy to severe AD [88]. 9 out of 12 modules were linked to disease progression. Of which, these modules were enriched for processes such as immune response and white matter. Significantly, when comparing AD and aging, two modules—one enriched for mitochondrial processes and the other enriched for synaptic plasticity—were preserved. However, it is important to note that many of these networks uncovered by WGCNA using microarray are cell type specific rather than process specific. This is because of the inherent whole-tissue nature of microarray data. Therefore, while WGCNA might allow us to investigate how genes are co-expressed in specific cell types, it might not be informative when it comes to how genes are co-regulated within a specific cell type.

Microarray-based network studies in mice have also provided new insights into AD pathogenesis and on the role played by specific cell types in the brain. Miller et al. investigated and showed conservation of gene co-expression networks across mice and human brains [89]. However, amongst the glial cells, microglia exhibit the greatest inter-species difference [89]. Another analysis across humans and AD mouse models, specifically 3xTg-AD and amyloid precursor protein (*App*)^NL-G-G/NL-G-G^ mice, showed an association between neuroinflammation and amyloidosis with multiple AD risk genes expressed in microglia [89,90]. Integrating human microarrays with mice microglia RNA-sequencing data (see next section), Mukherjee et al. performed a meta-analysis of gene networks using WGCNA in context of human aging and neurodegeneration, and mice aging [91,92]. Relevant “hub genes” have been identified in these network studies, including *TYROBP*, *PTPRC*, and *ITGB2* in humans, and *Trem2* in mice [91,92]. These findings not only highlight the conservation of AD risk genes across humans and mice, but also reinforce the critical regulatory role of TREM2/TYROBP interaction axis in microglia [36,93]. In addition, Mukherjee et al. confirmed conservation of gene networks across species by performing a module preservation analysis [91,92]. A module preservation statistic quantitates the extent to which the module is conserved (and detectable) across datasets. Therefore, in Mukherjee et al. a greater preservation statistic score would indicate conservation of particular gene network across species, in this case between mice and humans. Such an approach not only allowed inter-species comparisons, but also inter-experiment comparisons of gene networks, which are especially useful in the integration of multiple microarray experiments.

Highlighting the fundamental role and interplay between epigenomic and transcriptional regulation, microarray analysis on fluorescence-activated cell sorting (FACS)-sorted microglia identified histone deacetylases (HDACs), *Hdac1* and *Hdac2,* as key regulators of microglia homeostasis and development [94]. Other epigenetic elements, such as DNA methylation, acetylation, and chromatin structural states, can influence gene activity without changing the underlying DNA sequence [95]. Focusing on post-transcriptional gene regulation, miRNA microarray-based studies have also uncovered the role of miR-124 in microglia activation [96,97] and regulation of anti-inflammatory action and the peroxisome proliferator-activated receptor-gamma (PPARƔ) pathway [96]. Overall, even in the light of the subsequent advancements of transcriptomic sequencing technologies (see next section), microarray-based transcriptional network analyses have provided valuable new information on genes and functional regulatory processes in AD brain, and concurrent insights into the key role of microglia cells in AD pathogenesis.

### 3.3. RNA-Seq-Based Studies in AD

RNA-sequencing involves high-throughput sequencing of multifold RNA transcripts at once. Compared to microarray chips, RNA-seq has a much greater dynamic range, sensitivity, specificity, and does not require a priori knowledge of reference transcripts, allowing investigation and discovery of novel transcripts and splicing variants [75]. Starting from the latter, as reviewed by van Dam et al., exon-specific expression could be used to build gene co-splicing networks, [98], which can reduce bias that comes from a “splicing-naive” co-expression analysis [99]. This is especially important because gene splice variants can have different functions and impact splicing regulatory gene-modules [100]. Raj et al. investigated mRNA splicing in the dorsolateral prefrontal cortex (DLPFC) from 450 individuals to create a database of splicing quantitative trait loci or sQTLs (i.e., DNA sequence variants such as single nucleotide polymorphisms—SNPs—that affect the splicing of a gene transcript) [101]. On top of constructing co-splicing gene networks, Raj et al. found significant mRNA splicing dysfunction associated with aging brain, again highlighting the importance of studying information outside coding regions by RNA-seq in AD. In particular, known AD risk genes such as *CLU*, *PICALM*, and *PTK2B* were found to have unique splicing mechanisms [101], where *CLU* is a binding partner of *TREM2,* aiding microglial uptake of amyloid [102]. Furthermore, RNA-seq allows investigation of novel long intergenic noncoding RNAs (lincRNAs), which not only represent another layer of transcriptomic information, but are increasingly implicated in AD. Several studies now support the direct contribution of lincRNAs to AD pathogenesis by regulation of key functional processes, including neurogenesis, synaptic dysfunction, amyloid beta accumulation and neuroinflammation—reviewed in [103].

Importantly, Mostafavi et al. [104] used bulk RNA-sequencing from 478 individuals to derive networks of aging human frontal cortices. These networks were first constructed by calculating gene co-expression followed by linking the co-expression modules with AD traits using Bayesian analysis. This module-trait network (MTN) approach discovered networks associated with amyloid burden and cognitive impairment. In particular, the microglia-enriched module was associated with age, suggesting the importance of aging as a risk factor for AD pathogenesis. With respect to microglia, Zhang et al. created a mouse transcriptomic and splicing database of glia cells [105]. Notable examples of the genes identified as differentially spliced in microglia include *Clstn1* and *App* [105]. CLSTN1 (Calsyntenin 1) is known to play an important role in AD pathogenesis by regulating axonal transport of APP and subsequently production of amyloid [106]. Coupled with proteomics analysis, Johnson et al. used RNA-seq analysis in AD and control DLPFC to create a library of translated peptides, which was then analyzed for alternative splicing events [107]. Corroborating Raj et al. findings [101], *PTK2B* and *PICALM* were found to have many alternatively spliced isoforms, and splicing events were clustered in specific networks [107]. *PICALM* is a clathrin adaptor protein important for clearance of amyloid [108] while *PTK2B* is important for microglia (and other immune cells) activity [109]. With regards to lincRNAs, Magistri et al. conducted directional RNA-seq, which retains strand information to ascertain whether a transcript-associated read is from the positive or negative strand, on 4 LOAD and 4 control individuals, revealing upregulation 89 lincRNAs in AD with 72 of them not being previously reported [110]. Interestingly, upregulation of some lincRNAs in AD was specific to particular brain regions, whereas other lincRNAs were similarly expressed in cerebellum from AD patients and controls. Given the increasingly recognized role of lincRNAs in microglia phagocytosis [111], detailed mechanistic investigations of lincRNAs’ function in microglia is expected to provide additional and valuable insights into AD pathogenesis, beyond the role played by coding RNA transcripts.

Moving away from amyloid pathology in AD, Litvinchuk et al. analyzed RNA-seq data generated in human parahippocampal gyrus and hippocampus from control and tauopathy mouse model PS19 (an AD model initially developed to study immunotherapeutic targeting of tau pathology) [112,113]. Analysis of DE genes and integration with the TRANScription FACtor (TRANSFAC) database were used to pinpoint a dysregulated immune gene network in AD pathogenesis [112]. The *C3-C3ar* signaling was identified to underlie this network, which is also associated with viral-induced synapse loss [114], reactive astrocytes in AD mouse models [115], and synapse loss [116]. Importantly, the *C3ar* gene network contains *Spi1*, *Trem2*, and *Ms4a6a*, which mirrors the role of these genes in mouse models of amyloid build-up, suggesting a shared microglia response to tau and amyloid pathology. Of note, the signal transducer and activator of transcription 3 (Stat3), which might contribute to microglia activation [74,117], is a downstream target of the *C3ar* signaling network, and the downregulation of its downstream network ameliorates tau pathology [112]. Similarly, Wang et al. performed RNA-seq analysis in another murine model of tauopathy, rTg4510 [118] and found other members of the STAT protein family (Stat1, Stat6), and Rela to be mediating NF-kβ and cytokine during earlier stages of tau pathology [118]. Other microarray-based studies in the same rTg4510 mouse model corroborated the role of mediators of cellular immunity (STATs) and inflammation in AD pathogenesis [119].

Unsurprisingly, the development of RNA-seq (and its increasingly lower costs) was paralleled by a decline in popularity of microarrays [75]. Nonetheless, information obtained from the combined analysis of both RNA-seq and microarray datasets can be useful, especially given that both platforms provide a high throughput assessment of the transcriptome (but still RNA-seq studies are more expensive than microarray studies). For example, Zhang et al. constructed transcriptional networks from 1,647 postmortem brain tissues from LOAD patients and control individuals [120]. Zhang et al. utilized Bayesian inference to derive a microglia immune-module associated with clinical covariates relevant to AD [120]. Specifically, for the prefrontal cortex, 5 functional gene networks were found in the immune-module, related to (*i*) complement genes, (*ii*) cytokines, (*iii*) the Fc receptor system, (*iv*) the major histocompatibility complex, and (*v*) the toll-like receptor (TLR) signaling [120]—all of these pertain to the microglia’s role in amyloid phagocytosis, inflammatory signaling, and synaptic pruning [27]. Importantly, Zhang and colleagues associated increased network activity with AD severity, and showed TYROBP is a key regulator in a microglia module controlling phagocytosis [120], further implicating the TREM2/TYROBP signaling axis in disease pathogenesis. Although Zhang et al. also found common microglia regulators acting in different brain regions, analysis of discrete brain regions across the adult lifespan of the mouse, revealed that microglia have distinct region-dependent transcriptional identities [121], therefore suggesting the intriguing possibility of region- or context-specific regulators of microglia function in AD.

Transcriptomic data can also be integrated with “-omic” data sets outside proteomics. Novelly, Readhead et al. [122] integrated human transcriptomic data with proteomic, histopathological, and virome data to build multiscale networks in late onset AD (LOAD). The method of choice implemented here is probability causal network (PCN) analysis. Interestingly, Readhead et al. unveiled AD networks that are associated with viral activity, suggesting a role for viral infection in AD pathogenesis. Specifically, viral activity has been linked to APP metabolism with the involvement of various amyloid-related genes like *APBB2, BIN1,* and *BACE1*, which has been shown to be activated by the human herpesvirus 6A (HHV-6A). Overall, this demonstrates network analysis to be a powerful tool to integrate transcriptomic data with unconventional datasets (in this case the LOAD virome) for uncovering new insights into AD pathogenesis.

### 3.4. Single-Cell RNA-Sequencing (scRNA-sequencing) in AD

The advent of scRNA-seq technology has revolutionized transcriptomic analysis, especially in the context of development and disease. A key advantage of scRNA-seq is its ability to obtain cellular transcriptomes, allowing investigation of individual cells’ expression signature and detection of rare cell populations, including “intermediate” cell states [123,124,125]. Variants of scRNA-seq experimental designs come from the choice of platform (droplet-based vs well-based), extent of transcript coverage (full length vs 5’/3’ tagging), and additional features such as cell hashing, combinatorial barcoding, and isolation of nuclei-only RNA [124]. In conventional bulk RNA-seq analysis (i.e., in large populations of cells), the transcriptomic information is derived from a mixture of all cell types within the tissue sampled. Therefore, bulk RNA-seq cannot disentangle the confounding effect of changes in cell type proportion across samples [123]. This is further complicated by the stochastic nature of gene expression and cell type heterogeneity in complex tissues. On the other hand, scRNA-seq data has typically greater noise, level of dropouts (reflecting transcripts that failed to be detected due to technical limitations), low cell-capture efficiency, and technical variability compared to more established bulk RNA-seq [123,124,125]. Such limitations will be ameliorated and, despite these, scRNA-seq offers unprecedented resolution in elucidating cellular transcriptomics, which is increasingly helpful for interrogation of a tissue as complex and heterogeneous as the brain in AD.

Given the unique nature of scRNA-seq data mentioned above, computational tools and methods used in bulk RNA-seq analysis cannot be readily transferred to and implemented for scRNA-seq data analysis. For example, the high dropout rate in scRNA-seq data can result in many undetected transcripts, which require the implementation of specific statistical models (e.g., negative binomial model with zero inflation) for accurate handling of this kind of “missing” data [126]. As for DE analysis in single-cell data, gene network inference from scRNA-seq data is therefore substantially different from data generated by microarray or bulk RNA-seq. The primary, distinctive advantage of scRNA-seq is the ability to assess complete transcriptomes in high number of cells (typically in the order of thousands of cells), where each cell can theoretically be treated as an individual sample-replicate [127]. Therefore, this additional level of biological variability and information entails more complex descriptions of gene co-expression networks [127]. Todorov et al. delineated three main network outputs from scRNA-seq data—differential, dynamic, and profile-specific [127]. Briefly, differential network output relates to networks derived from differential expression analysis between specific cell groups, e.g., between microglia and neurons [128]. Dynamic network output leverages on “pseudotime”, which is a scRNA-seq-specific analysis that allow to order cells by their gene expression profile(s), reflecting either a biological trajectory such as developmental time-points or varying cellular states (e.g., control vs stimulated) [129]. The network activity is subsequently mapped across this pseudotime trajectory, yielding a dynamic network output [127,130,131]. Profile-specific network outputs relate to individual networks that are unique to a distinctive expression profile, which could be derived from a specific cell type/group (e.g., microglia-specific), an a priori cell group of interest (e.g., cells from disease brain), or from individual cells (e.g., dopaminergic neurons) [127,132].

Single cell RNA-seq network analysis in AD mouse models, while still in its infancy, have already yielded new insights into the role of microglia in AD pathogenesis. For instance, Keren-Shaul et al. identified a Trem2-dependent microglial signature termed the damage-associated microglia (DAM) which is conserved across aging, AD, and amyotrophic lateral sclerosis (ALS) [133], supporting the key role of Trem2 that was hitherto suggested by microarray-based and RNA-seq analyses [120]. Subsequently, Rangaraju et al. further dissected the DAM signature separately into pro-inflammatory and anti-inflammatory modules using WGCNA, and posited that the correlation of pro-inflammatory DAM component with neuropathology might account for the damaging role of DAM [62]. Knocking out *Trem2* in 5xFAD mice causes dampening of both pro- and anti-inflammatory networks [120]. Liu et al. described an anti-inflammatory *Trem2* network encompassing known AD risk genes such as *SPI1*, *MS4A6A*, and *CD33*, which responds to amyloid build-up [134]. Therefore, it has also been suggested that the activation of this anti-inflammatory gene network can occur in response to amyloid build-up as a protective mechanism. Conversely, activation of pro-inflammatory networks might exacerbate AD pathology by releasing cytokines and by pruning of synapses. To further investigate the key regulatory of Trem2, Iacono et al. employed a novel correlation measure tailored to analyze single-cell data correlation metrics to infer regulatory networks [135]. Comparing microglia cells from different disease stages and with different Trem2 “dosage” (i.e., Trem2^+/+^ vs Trem2^-/-^), Iacono et al. observed that AD gene networks have fewer connections compared to healthy controls, showing extensive rewiring of regulatory networks in AD microglia. In particular, genes that lost connectivity within the network significantly overlap with Trem2-dependent genes in monocytes [135], suggesting that Trem2 might be important in restricting AD pathogenesis. Figure 2 summarizes key findings of the role of Trem2 in microglia with respect to AD pathogenesis.

Specific to single cell transcriptomics, Single Cell Regulatory Network Inference and Clustering (SCENIC) is a scRNA-seq profile-specific network inference method which uses a random forest classifier to identify TFs followed by TF binding motif enrichment analysis to link downstream TF-targets [132]. Therefore, SCENIC leverages also on DNA sequence (i.e., TF binding sites) analysis to reconstruct regulatory networks. Aibar et al. employed SCENIC on scRNA-seq data consisting of 3005 brain cells isolated from hippocampus and cortices of juveniles mice, and identified many key network regulators in microglia such as PU.1/Ets family of TFs, Nfkb, Irf, and AP-1/Maf [132,137]. Interestingly, in comparison with published microglia “activation” gene signatures in AD [138], the SCENIC-inferred microglia network was found to be strongly upregulated in AD pathogenesis [132]. Similarly, using SCENIC on the mouse single cell atlas, Suo et al. identified an immune gene network consisting of *Mafb*, *Irf2*, and *Nfkb1* to be highly expressed in microglia [139]. In particular, *MAFB* is a critical microglia master regulator involved in microglia development and immune regulation in humans [136]. Importantly, scRNA-seq also allows for transcriptomic interrogation of rare cell populations which otherwise would not be accessible by either microarray and bulk RNA-seq. For example, Tay et al. performed scRNA-seq of microglia from a mouse model of unilateral facial nerve axotomy, which recapitulates acute neurodegeneration, and found that microglia regulatory profiles are preserved between acute and chronic neurodegeneration [140]. Some conserved transcriptional profiles include genes regulating antigen presentation, response to interferons, TGFβ signaling, and cell migration [140]. Hammon et al. further investigated microglia transcriptomic changes not only in pathology, but across the trajectory from young to old mice, thereby modelling aging, which is an established risk factor in AD [141]. At least 9 microglia subpopulations were identified along the mouse lifespan, and, intriguingly, genes activated during disease were likely also expressed in development [141]. Therefore, scRNA-seq unravels rare microglia sub-populations in different stages of aging and AD pathogenesis, providing a new opening to the study of neurodegeneration in terms of its acute, chronic, and resolution stages.

Most scRNA-seq studies of AD have been carried out in mice which primarily model familiar AD and, being based in rodents, these are unlikely to reproduce accurately “human brain aging” as the major AD risk factor. Recently, Mathys et al. performed snRNA-seq on post-mortem prefrontal cortices from 48 individuals with different degrees of AD pathology [142] and utilized self-organizing maps (SOMs) to identify gene sets associated with various clinical phenotypes [142]. SOMs are unsupervised neural networks used to cluster noisy data (such as scRNA-seq data) and group related genes into discrete gene networks and modules [143]. In Mathys et al., the gene modules were also termed “gene-trait correlation modules” because they correlate with various pathological or clinical traits such as neurofibrillary tangle burden/density, amyloid levels, and global cognitive function. In particular, two gene-trait correlation modules (termed M6 and M7) were highlighted as these were positively correlated with microglia and pathological traits, and contained AD risk factors such as *APOE*, *TREM2*, *MEF2C*, and *PICALM*. The M7 gene module was also enriched for genes related to immune processes and amyloid clearance [142]. Importantly, Mathys et al. also investigated sex-specific effects in AD at the single cell level, revealing transcriptional responses that were substantially different between sexes in several cell types (e.g., oligodendrocytes). This exemplifies how scRNA-seq and transcriptomic network analysis can be integrated with clinical and pathological phenotypes in humans to reveal molecular and cellular changes in the activity of specific networks and genes in AD pathogenesis.

Despite the intrinsic differences between microarray, RNA-seq, and scRNA-seq, integrating transcriptomic data from all technologies can be useful. For example, Patir et al. combined analyses of 15 datasets generated using the 3 platforms in order to derive a “core” human microglia gene network [144]. The microglia network was constructed using network analysis tool Graphia Professional [145], which involves the generation of a gene co-expression network and the derivation of gene clusters by Markov clustering algorithm. Overall, 249 genes, including known microglia markers *CD68*, *P2RY12*, and *TMEM119* were found within a core transcriptional network, which was conserved across species (human and mouse), platforms, and brain regions [143]. In addition, Patir et al. found that microglia numbers change accordingly with different levels of inflammation and age-related atrophy, reflecting known region-dependent alterations in AD. For example, the vulnerable superior frontal gyrus showed a substantial increase in microglia numbers, while the relatively AD-resistant post-central gyrus exhibited little change in microglia numbers. Furthermore, co-expression network analysis further refined 165 microglia-associated genes (MAGs), which were not inside the “core” gene network, and found that these MAGs were enriched in lipid regulation, showing that microglia are implicated not only in inflammation, but also in metabolic homeostasis in the context of AD [144]. This study illustrates how the unbiased identification of different networks can provide insights into distinct and specific molecular processes and pathways, which are otherwise difficult to disentangle from the analysis of transcriptional signatures and DE genes alone.

Similarly, Friedman et al. performed a meta-analysis of CNS myeloid cells across 69 conditions including mouse models of neurodegeneration, aging, ischemia, infection, and tumor proliferation, identifying various gene modules related to neurodegeneration and interferon-related processes [146]. These co-regulated gene modules were derived by performing DE and hierarchical clustering analysis, starting from a common set of genes commonly expressed across all conditions tested. The key neurodegeneration gene module included 4 transcription factors—*Bhlhe40*, *Rxrg*, *Hif1a*, and *Mitf*. Other studies have implicated *Hif1a* in AD pathogenesis [25,147]. In particular, Baik et al. posited that microglia activation requires *Hif1a* to switch on glycolysis, while chronic neurodegeneration results in the microglia metabolic breakdown [25,147]. The neurodegeneration module also contains cathepsins, which reflect lysosomal activity and *Apoe*, the key risk gene for AD [146]. In addition, this module was also detected in the P301L genetic model of tauopathy, suggesting the importance of microglia not only for amyloid pathology, but for tau pathology. A summary of the key biological insights into AD pathology by transcriptional network analysis which have been discussed here, is reported in Table 1.

## 4. Further Applications of Transcriptomic Gene Networks in AD

Typically, transcriptional gene networks provide information of biological processes associated with the disease trajectory in a given tissue or cell-type. These networks could also elucidate AD pathogenesis and potentially be used to suggest drug targets. This inference is boosted by additional integration with orthogonal “*-omics*” datasets and/or sources of biological information (e.g., curated knowledge-based databases, functional gene ontologies and annotations, and drug transcriptional-perturbation responses database).For example, Vargas et al. incorporated the identified gene regulatory networks with the Connectivity Map Analysis (CMap) database, and identified drugs capable of perturbing gene networks in AD, showing the utility of gene network analysis for drug repositioning [64]. Briefly, CMap contains a transcriptomic database of drug-induced cell perturbations, providing a reference for comparison with specific disease-associated gene signatures and networks [149]. This allows identification of drugs which might potentially exacerbate or reverse the input gene signature/network. Vargas et al. used published gene expression profile data (Gene Expression Omnibus (GEO) database, https://www.ncbi.nlm.nih.gov/geo) and employed the ARACNe algorithm to identify TF-driven regulatory networks in human hippocampus, which were further connected to disease using AD case-control studies. These TF-driven regulatory networks were used to predict drugs by CMap, which yielded six FDA-approved drugs (Cefuroxime, Cyproterone, Dydrogesterone, Metrizamide, Trimethadione, and Vorinostat) predicted to potentially counteract AD pathogenesis. In particular, Vorinostat is a histone deacetylase inhibitor (HDACi) which has not only been shown to protect against amyloid toxicity [150], but is also currently under clinical trial for memory improvement (www.clinicaltrials.gov). With regards to microglia in particular, vorinostat can inhibit the release of cytokines after exposure to lipopolysaccharide, which is a gram-negative endotoxin and pro-inflammatory stimulus [151]. This corroborates findings of independent high throughput drug screenings, which showed that vorinostat can reduce PU.1 expression in human microglia [152], potentially limiting neuroinflammation and retarding AD pathogenesis. In addition, Vargas et al. identified the AD master regulator *PARK2*, which has been shown to be associated with NLRP3 inflammasome activation in microglia [153], and further incorporated CMap data to discover drugs capable of perturbing identified networks, highlighting the utility of gene network analysis in drug repositioning.

Following the central dogma of molecular genetics, the flow of information from mRNA to protein is influenced and regulated at multiple levels, including translation, splicing, and polypeptide processing. Therefore, proteomics data has been widely used to further refine networks inferred from transcriptomic analyses. For example, Canchi et al. integrated mRNA expression profiles from prefrontal cortices from 414 AD and control individuals with brain tissue-specific protein interactomes [154]. This was achieved by cross-matching the many differentially expressed genes between AD and controls with the Genome-Scale Integrated Analysis of Networks in Tissue (GIANT)—a database of human tissue-specific networks [155]. This allowed Canchi et al. to find downregulation of synaptic signaling, metabolism, cell survival and proliferation, and immune responses, and an upregulation of several novel processes including small guanosine-triphosphate (GTPase) signaling. The authors further discovered EGR3, TGIF1, SP1, and BPTF as TFs associated with AD pathogenesis. EGR3 and EGR1 have been implicated in short-term memory [136] and maintenance of long-term potentiation (LTP) respectively [156]. Importantly, EGR3 was not previously associated with AD and, thereby, Canchi et al. exemplified the utility of integrating proteomic and transcriptomic data to suggest new TFs involved in AD pathogenesis. Similarly, Marttinen et al. combined microarray and phosphoproteomic data from human temporal cortices in a “multi-omics” approach and identified mitochondrial and synaptic dysfunction as the earliest pathological processes occurring in AD pathogenesis [157]. This approach allowed for the reconstruction of the protein–protein interaction (PPI) networks which uncovered relationships between diverse biological processes altered at the different stages of AD pathogenesis, and revealed increase in kinase activities in AD, involving as CDK5, GSK3β, PRKACA, PKN1, and MAPK12.

Overall, network analysis aids in the investigation of complex diseases like AD by moving away from a reductionist approach. By interrogating the interplay of genes, network analysis not only implicates the key genes driving various pathological processes in AD, but also provides a compendium of target genes for possible therapeutic perturbation. Furthermore, network analysis can further be refined to discover cellular subtype-specific networks and incorporate molecular information. For example, Roussarie et al. [158] made use of bacTRAP and human genomics data to understand context-specific neuronal networks in AD. A key development in the future would be the integration and utilization of the wealth of networks in AD literature. This “harmonization” of network information would be useful in interpreting networks in different mouse models, translating findings from mice to humans, and dissecting networks built from different stages of AD pathogenesis.

## 5. Concluding Remarks

During the last decade, transcriptomic network analyses have demonstrated the microglia’s functional role in inflammation [133], synaptic pruning [159], neuronal damage [160], metabolic changes [161], and amyloid clearance in AD [162]. After the pioneer microarray-based transcriptomic studies, followed by improved RNA-seq-based studies, scRNA-seq has provided the most significant breakthrough in the field. This relates to the opportunity of high-content and cell-resolved transcriptomic analyses in otherwise complex and heterogeneous tissues—which has been crucial to decode pathological processes occurring in specific cell populations in the AD brain. Integration of auxiliary regulatory information at single cell level aided a more robust inference of gene regulatory networks underlying AD pathogenesis, which further highlighted the primary role played by microglia as well as their interplay with other resident brain cells [24,163,164]. Moving forward, spatially and temporally resolved scRNA-seq analyses of microglia gene networks are likely to provide further elucidation of region-specific microglia networks and altered networks’ dynamics associated with AD pathogenesis. This is especially important given the association between the spread and load of AD pathological correlates (β-amyloid and tau) across different brain structures with AD severity. In the meanwhile, researchers have the opportunity to take advantage of increasingly available single cell transcriptomic data (e.g., https://www.humancellatlas.org, http://portal.brain-map.org/ ), and integrate these with orthogonal “*-omics*” data types (e.g., scATAC-seq, scChIP–seq, scTHS-seq, scDNA-methylation profiling, scProteomics, scMitochondrial DNA-seq) for accurate network construction at single-cell resolution. This development of “multimodal” (or “multi-omics”) profiling strategies at the single cell level will likely prompt the development of innovative statistical and computational approaches to analyze all these data types in a unified framework [165]. The expectation here is to move beyond the transcriptome-centric gene network view of AD and acquire a comprehensive multi-layered representation of the cell’s dynamic state during disease progression and in pathogenesis.

## Figures and Tables

**Figure 1 genes-10-00798-f001:**
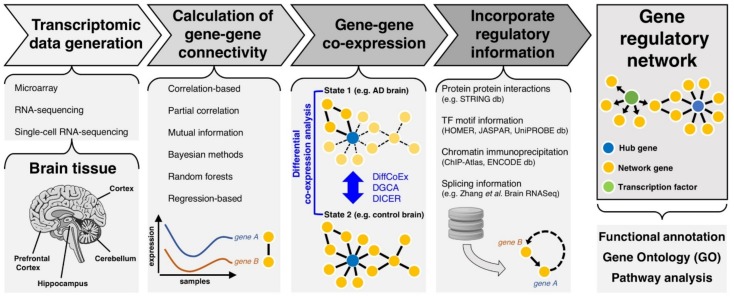
Schematic overview showing the workflow for gene network inference, from transcriptomic data generation to integration of regulatory information to assess gene regulatory networks.

**Figure 2 genes-10-00798-f002:**
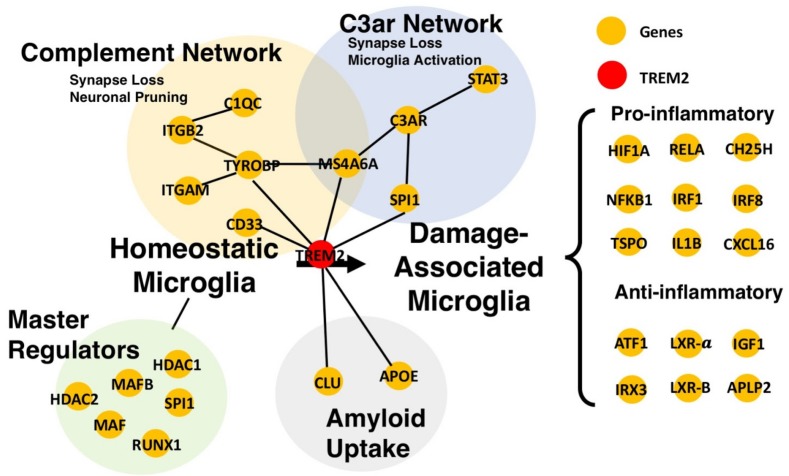
Network representation of Triggering Receptor Expressed on Myeloid Cells 2 (TREM2) and its interacting partners in the context of the switch from homeostatic microglia to damage-associated microglia (DAM). Only representative genes are shown in the network. Hence, the genes included in this TREM2-network are neither comprehensive nor exhaustive of the processes, transcriptional networks or signature they belong to. However, several transcriptional gene network studies demonstrated the role of TREM2 at the center of a complex interplay between cellular processes (e.g., synapse loss and neuronal pruning) and multifold genetic and epigenetic regulators, which account for the dynamic nature and functions of microglia in AD. Representative genes shown are taken from Rangaraju et al. (2018) [62]*,* Litvinchuk et al. (2018) [112]*,* Keren-Shaul et al. (2017) [133], Matcovitch-Natan et al. (2016) [136], and Zhang et al. (2013) [120].

**Table 1 genes-10-00798-t001:** Overview of the relevant biological insights into Alzheimer’s disease (AD) pathology obtained by transcriptional gene network analysis, which was performed using different transcriptomic technologies, species, tissue types and network inference approaches.

Insights into AD Pathology	Technology	Species - Tissue type	Network Inference Method	References
-Upregulation of neural signaling elements and pro-inflammatory elements	Microarray	*Human* - Hippocampal CA1	Cluster analysis	[78]
-CD4, DCN, and IL8 extracellular ligands linked to disease initiation -Implication of miRNA-networks in AD pathogenesis	Microarray	*Human* - Hippocampal CA1 - Entorhinal Cortex	Network Topology Analysis	[82]
-*PSEN1* is strongly associated with myelin proteins -Conservation of modules for metabolism and synaptic plasticity conserved between AD and aging	Microarray	*Human* - Hippocampal CA1	WGCNA	[88]
-*Apoe* implicated as a general aging gene and associated with syndromic learning impairment	Microarray	*Rat* - Hippocampus	Ingenuity Pathway Analysis (IPA)	[83]
-*Cdk5r1, Dlg3, Kcnab2,* and *Mapk1,* and *Camk1g* identified as hub network genes and associated with ion signaling and learning	Microarray	*Rat* - Hippocampus	WGCNA	[86]
-*TYROBP, PTPRC, ITGB2,* and *Trem2* identified as “hub” genes in AD gene networks in humans and mice, respectively -Conservation of genes across humans and mice AD	Microarray RNA-Seq	*Human* - Prefrontal Cortex - Substantia Nigra *Mouse* - Hippocampus	WGCNA	[91,92]
-Role of splicing quantitative trait loci and co-splicing gene networks in AD -*CLU, PICALM,* and *PTK2B* show unique splicing mechanisms in AD	RNA-Seq	*Human* - Dorsolateral Prefrontal Cortex	WGCNA GeNets	[101,148]
-PARK2 associated with NLRP3 inflammasome in microglia activation - 6 FDA-approved drugs (Cefuroxime, Cyproterone, Dydrogesterone, Metrizamide, Trimethadione, and Vorinostat) predicted to modulate microglia master regulators	Microarray	*Human* - Hippocampus	ARACNE (Algorithm for the Reconstruction of Accurate Cellular Networks)	[63,64]
-Role of splicing gene networks of microglia in AD, and identification of *App* and *Clstn1* as differentially spliced -Splicing occurs differentially across different cell types in AD brain	RNA-Seq	*Mouse* - Cortex	WGCNA	[105]
-Alternative exon-exon junction splicing in AD brain	RNA-Seq	*Human* - Dorsolateral Prefrontal Cortex	WGCNA	[107]
-*C3-C3ar* signaling associated with viral-synapse loss and reactive astrocytes, and its downregulation reduces tau pathology -*Spi1, Trem2,* and *Ms4a6a* part of *C3ar* gene network	RNA-Seq	*Human* - Parahippocampal Gyrus *Mice* - Hippocampus	Correlation-based	[112]
-TYROBP identified as a key regulator in a microglia module controlling phagocytosis -Different brain regions have different regulators -Functional gene networks identified for prefrontal cortex including complements and cytokine networks	RNA-Seq	*Human* - Dorsolateral Prefrontal Cortex - Visual Cortex - Cerebellum	Module Differential Connectivity (MDC) Causal probabilistic Bayesian	[120]
-Dissection of Damage-Associated Microglia(DAM) into pro-inflammatory and anti-inflammatory modules	Microarray Nanostring	*Mouse* - Whole brain	WGCNA	[62]
-*Trem2* network becomes sparser with AD pathogenesis; specifically, genes that lost connectivity overlap with *Trem1*-dependent genes in monocytes	scRNA-seq	*Mouse* - Cortex - Hippocampus	Correlation-based	[135]
-Identification of master microglia gene regulators including PU.1/Ets family of TFs, Nfkb, Irf, and AP-1/Maf -Microglia gene network upregulation in AD	scRNA-seq	*Mouse* - Cortex - Hippocampus	SCENIC (Single Cell Regulatory Network Inference and Clustering)	[132]
-Identification of microglia immune network containing *Mafb*, *Irf2*, and *Nfkb1*	scRNA-seq	*Mouse* - Brain	SCENIC	[139]
-*APOE, TREM2, MEF2C,* and *PICALM* implicated in microglia gene-trait correlation modules	scRNA-seq	*Human* - Prefrontal Cortex	SOM (Self Organizing Maps)	[142]

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
