# Peer review of "Transcriptional Networks of Microglia in Alzheimer’s Disease and Insights into Pathogenesis"

_genes, 2019, doi:10.3390/genes10100798_

Round 1

Reviewer 1 Report

The manuscript titled “Transcriptional Networks of Microglia in Alzheimer’s Disease and Insights into Pathogenesis” by Gabriel Chew and Enrico Petretto focuses on benefits vs limitations of different network analysis methods.

Various methods of gene and network analysis and their pros and cons are comprehensively described. Authors go into significant detail for each of the methods described in the manuscript. The description of the methods and general applications and limitations is very well written.

Key criticism: While the title, abstract and introduction focus predominantly on microglia and AD, the article focuses for most part on the details of general methodology for network analysis and then for a small part, focuses on the application to AD, followed by microglia. While the methods are well described, their relevance to AD and microglia seems to be tangent to the core message of the paper. Suggestion to the authors is to make AD and microglia as central/core part of the review and describe the methods within the context of the disease.

Additionally, in Section 4 authors should describe the overall impact of the network analysis on advancing knowledge of microglial biology in AD and explain how these methods can help address gaps in our current understanding of the role of microglia in the pathobiology of this neurodegenerative disorder.

Minor corrections: Typos: line 35- trigger should be triggers. Line 65: “a” should be removed from “for a microglia’s key role in AD.”

Reviewer 2 Report

The manuscript entitled "Transcriptional Networks of Microglia in Alzheimer's Disease and Insights into Pathogenesis" is a review that gives a very comprehensive overview of network-based approaches for the study of microglia in Alzheimer's. They first introduce networks, what kind of data they rely on, and how they can be constructed. They then present a very detailed historical overview of transcriptional studies, especially in the field of AD. They then give an overview of the network analyses that were performed using these transcriptional data, to model microglia function in AD.

Overall the manuscript is a very good resource for non-bioinformaticist experimentalists, who could be a bit lost in front of the proliferation of network approaches, as well as for bioinformaticists. Conceptually difficult approaches are presented with clarity, and the manuscript seems to be very comprehensive, to the best of my knowledge (at the price of looking a little list-ey, at times), except for a few important references, which should not be left out, as they are extremely important for the field:

the first (if you exclude the very minimal clustering of reference 78) publication doing network analysis on AD: Miller, Oldham, Geschwind, 2008, J Neurosci Readhead et al., 2018, Neuron, with networks very carefully constructed, focusing on specific relevant brain regions, and finding network drivers for normal and AD brains Mostafavi et al., 2018, Nat Neurosci, for their use of the landmark ROSMAP dataset, and their exciting attempt to correlate different modules with different endophenotypes. It is the direction where network analysis should do, informing the precise molecular dissection of the disease Roussarie et al., 2018, Biorxiv, for the attempt to use molecular signature to extract cell-type specific functional networks. Even if that preprint focuses on neurons, it could serve as a template for microglial cell specific networks.

On the other hand, the authors probably spend too much time giving on overview of transcriptional techniques. Obviously these techniques are crucial for the generation of the data that will then be used for the networks, it is still not necessarily the most important part of the review. A whole history of gene expression measurement techniques as given in part 3. is probably not indispensable, or at least could be made a bit more concise.

Similarly the paragraph starting line 62 is very detailed on microglia, and makes one lose the focus of networks.

It might be a good idea to clarify a point that distinguishes the older WGCNA-based bulk analyses from any attempt to construct networks at the cell-type level: a lot of the modules found in WGCNA approach relate to a cell type more than to a specific process inside the cell, because with bulk sequencing a great determinant of gene-gene co-expression will be the expression in the same cell type. Cell-type specific processes associated with very large changes in gene expression will also be visible, but some processes that are more subtle are probably going to be missed. It is important to highlight that point, because very often we thus end up not studying how genes are co-regulated within a cell but how genes are co-expressed in similar cell types. Only approaches like single-cell/single-nucleus or cell-type specific functional networks could get to the subcellular transcriptome architecture.

Minor points:

line 75-76: I can't think of any process where there wouldn't be a role for gene-gene interactions. Please find a better formulation for the description of this reference. line 113: "the gene-gene connectivity is typically described as gene co-expression" is a weird formulation. "is typically based on"? p5, please correct the appearance of the font in figure 1. line 290: studying splicing is not studying non-coding information. Maybe the authors mean the study of sQTLs that are not necessary in coding regions?  the reference of 103 (lines 299 to 304) is a bit weird since they do not make any use of WGCNA - they conduct it but do not comment it. In general, greater attention should be placed to focus the review on network approaches and not just transcriptomic approaches. To my knowledge, reference 105 was also contested recently. Since it is outside of the scope of this review, I think it should be removed. line 299 should be glial cells reference 109 is not analyzing cell-type specific lincRNAs, so it should not be in the paragraph about microglia. Additionally, they do not really perform a network analysis on lincRNAs. In my opinion lines 310 to 318 could be removed. Lines 441-443 should either be removed or made more precise, saying what hypotheses could be corroborated or disproven. line 472: "the study uncovered AD genes not previously detected in animal models, including C1QB and CD14". I know the formulation comes from Mathys and al. But it is imprecise and wrong. Imprecise because it gives the imprecision the genes are genetically linked with AD. And wrong because because both C1qb and CD14 are absolutely detectable in the mouse, in single-nuclei sequencing, as well as in the Ben Barres database (at extremely high levels actually). In addition, the paper's findings about sex-specific processes are not really relying on network approaches, more on just regular differential gene expression approaches. Their work with network is pretty minimal. table 1:no  need to spell out what WGCNA stands for on every row  everywhere across the manuscript: to my knowledge, every time "mouse" is used as qualifier, it should be invariant: mouse model, mouse ageing, mouse single cell atlas, etc... in my opinion, publications using IPA based networks, which are relatively cryptic in terms of how they are constructed, should not necessarily be mentioned, as it doesn't seem to be really best practice to start a network project with IPA as the only tool.
